# Can green credit policies improve the digital transformation of heavily polluting enterprises: A quasi-natural experiment based on difference-in-differences

**Xuan Zhou[1], Dejia Yuan[2]\*, Zhengwei Geng[3]**

1 School of Economics and Management, North University of China, Taiyuan, Shanxi, China, 2 School of Economics and Management, North University of China, Guiyang, Guizhou, China, 3 School of Economics and Management, North University of China, Xingtai, Hebei Province, China

\* 1826898211@qq.com

**Data Availability Statement:** All relevant data are within the manuscript and its Supporting Information files.

## Abstract

The digital transformation of the manufacturing industry is closely linked to green credit policies, which jointly promote the development of the manufacturing industry towards a more environmentally friendly, efficient and sustainable development. Based on the research sample of China's manufacturing A-share listed companies from 2008 to 2022, this paper uses the difference-in- differences (DID) method to analyze the impact of green credit policies on the digital transformation of heavily polluting enterprises. The results show that green credit policies significantly inhibit the digital transformation of heavily polluting enterprises. In terms of the adjustment mechanism, the R&D investment of enterprises and the financial background of senior executives have weakened the inhibitory effect of green credit policies on the digital transformation of heavily polluting enterprises. When the R&D investment is low, the inhibitory effect of the policy is more significant, but with the increase of R&D investment, the inhibitory effect of the policy gradually weakens, indicating that there is a substitution relationship between the two. Enterprises with senior financial expertise have a deeper understanding of financial feasibility and benefit analysis, and are more receptive to the high-risk investment of digital transformation, while their financial network resources can help broaden financing channels, reduce financing constraints, and further reduce the financial difficulty of digital transformation. In addition, the green credit policy has a stronger inhibitory effect on the digital transformation of non-state-owned enterprises and enterprises that do not hold bank shares. The conclusions of this paper are expected to provide some policy implications for the subsequent green credit policies in promoting the digital transformation of the manufacturing industry.

## Introduction

Today, as the world experiences rapid digital technology and rising environmental issues, the challenges facing businesses are more complex and urgent. The frontier of digital technology

**Funding:** The author(s) received no specific funding for this work.

has not only changed the business landscape, but also redefined the position of enterprises in global competition. At the same time, global environmental problems, such as climate change and resource depletion, are threatening the sustainable development of enterprises. As a result, digital transformation and environmental protection, as the two major themes that will lead the future development, are gradually becoming the core elements shaping the corporate strategy. On the one hand, driven by the current wave of digitization, the manufacturing industry is undergoing a profound change, and digital transformation has become a strategic choice for enterprises in meeting future challenges and seizing opportunities, as a strategic initiative integrating advanced technology and innovative thinking, which is leading the manufacturing industry into a new era. The digitalization of the manufacturing industry is a product of internal and external environmental factors [1], which has a significant impact on the production process and management process of enterprises, which not only changes the traditional production methods, but also leads to a major transformation of enterprise management, marketing, product innovation and other levels. On the other hand, environmental protection is of great importance in today's global economy. Manufacturing companies must comply with increasingly stringent environmental regulations and standards, which is not only a social responsibility for enterprises, but also an important way to achieve sustainable development. Environmental requirements are driving companies to innovate in technology and business models, and to explore new development opportunities. Through R&D and application of environmental protection technologies, enterprises can develop new products and services, and open up new markets, which can not only enable the manufacturing industry to meet the regulatory requirements of the green environment and social expectations, but also improve resource utilization efficiency, reduce operational risks, enhance market competitiveness, and explore innovation and development opportunities. Driven by digitalization and environmental protection, manufacturing enterprises should integrate environmental protection into their strategic planning, promote green transformation, and achieve a win-win situation of economic and environmental benefits.

As a financial instrument to encourage environmental initiatives, green credit policies provide a new source of funding for companies, and by rewarding environmental measures, they may play a key role in driving companies to participate more actively in the process of digital transformation. From the perspective of capital, the digital transformation of the manufacturing industry requires huge financial support, and this policy may provide enterprises with a way of sustainable financing, which is expected to alleviate the huge financial pressure they may face in the digital transformation. From the perspective of incentive mechanism, green credit policies may also become a driving force for enterprises to take the initiative to move towards digital transformation. With the emphasis of the government and society on environmental responsibility, enterprises are expected to obtain more favorable green credit terms by adopting digital technologies to improve the efficiency of production processes, reduce resource waste, and reduce environmental emissions.

However, while green credit policies may encourage firms to invest more in environmentally friendly technologies in the short term, their specific impact on long-term technological innovation by firms, especially digital transformation, which requires a significant investment of capital and time to bear fruit, remains an area of challenge and unanswered questions. Firstly, the complexity of digital transformation is reflected in the fact that it is not just a technological update, but a comprehensive organizational change. It involves adjustments to company culture, employee training, and the integration of new technologies on a number of levels, all of which take time and effort to change. While short-term green credit policies incentives may push companies to make initial investments in environmentally friendly technologies, to achieve digital transformation in the true sense of the word, companies need longer-

term plans and commitments. Second, investments in digital transformation are not permanent and require a continuous injection of capital at different stages. The incentives provided by green credit policies in the short term may not be able to meet the funding needs for the entire transformation cycle. Enterprises may receive some financial support in the initial stage, but the scale and frequency of financial investment may gradually increase as the project deepens and expands. In summary, enterprises must explore the relationship between green finance and digital transformation more actively while pursuing sustainable development. Especially for those heavily polluting enterprises, digital transformation is not only a need to enhance their competitiveness, but also an urgent requirement to fulfill their social responsibility. In this context, whether green credit policy can become a catalyst to promote the accelerated digital transformation of heavily polluting enterprises is a question that deserves in-depth exploration.

Currently, academics have conducted a lot of research on manufacturing digital transformation and green credit policies respectively. On the one hand, studies have shown that digital transformation helps alleviate the information asymmetry between investors and enterprises, and between enterprises and product supply and demand markets, enabling investors to more accurately assess the value and potential of enterprises [2]. At the same time, the information asymmetry between enterprises and product supply and demand markets has also been alleviated to a certain extent, which leads to more efficient operation of the market. Enterprise digital transformation through digital technology, enterprises can more easily access to financing channels and financing information, improve the flexibility and efficiency of financing, can ease the enterprise financing constraints and reduce the cost of financing, which provides a wider range of financial support for the development and expansion of enterprises, and helps to promote the innovation and upgrading of the manufacturing industry [3]. In addition, digital transformation can significantly improve the innovation efficiency of enterprises, especially green innovation [4]. Digital knowledge management (KM) has a significant positive impact on technological innovation, mainly through absorptive capacity, adaptive capacity and innovative capacity [5]. Meanwhile, the digital transformation of high-tech industries has a positive effect on both technological innovation and achievement transformation [6]. On the other hand, in terms of green credit policies, the introduction of the Green Credit Guidelines in 2012 marked the official implementation of green credit policies, which is the core of China's green credit policies system and an important perspective for many scholars to study [7]. However, most current studies show that the implementation effect of green credit policies is not satisfactory [8]. On the one hand, green credit policies will inhibit bank loans and long-term financing of heavy polluting enterprises through financing constraint theory and financing cost theory [9], and significantly reduce long-term bank loans of heavy polluting enterprises [10]. On the other hand, the green credit policy significantly inhibits the level of technological innovation of heavy polluters [11]. Maybe the policy will improve the sustainable development of enterprises in the short term, but it has no long-term effect [12] and promotes poorly managed zombie enterprises [13].

In summary, digital transformation and green credit policies are key factors in the process of high-quality development of the manufacturing industry in terms of technological innovation, transformation and upgrading. At present, there is a large number of literatures on the digital transformation of the manufacturing industry and green credit policies, but few studies combine the two to explore the relationship between green credit policies and the digital transformation of the manufacturing industry. Therefore, the marginal contributions of this paper may be: Firstly, the uniqueness of the research: This paper may be the first time to deeply explore the relationship between digital transformation in the manufacturing industry and green credit policies, combining these two key areas for research. This research is unique in

that it connects the two key themes of digital transformation and environmental policies, filling a gap in the existing literature and providing a new research perspective for the academic community. Secondly, the importance of research to academia and practice: This paper fills the gap in the academic understanding of the relationship between digital transformation and green development in the manufacturing industry, and provides new ideas and methods for solving problems in this field. At the same time, the research results of this paper are of great significance for practice, which can provide useful reference suggestions for China's green credit policies formulation and digital transformation of the manufacturing industry, promote the sustainable development of the manufacturing industry, and promote the development of China's economy in a greener and more innovative direction. Thirdly, the theoretical and empirical contributions of the research: By exploring the impact mechanism of green credit policies on the digital transformation of the manufacturing industry, this paper expands the existing theoretical framework and provides new ideas and perspectives for theoretical research. Besides, this paper provides new empirical evidence based on empirical data, deepens the understanding of the mechanism of green credit policies in the process of digital transformation, and provides strong support for practice in related fields. Fourthly, the potential impact of the research: The research results of this paper are expected to have a profound impact on policy-making and practice. By proposing more effective green credit policies to promote the sustainable development of the manufacturing industry, this paper will help guide the government and enterprises to better formulate policies and strategies, promote the development of China's manufacturing industry in a more digital, green and sustainable direction, and contribute to the realization of high-quality economic development.

## Materials and methods

### Theoretical analysis and research hypothesis

Digital transformation typically requires large-scale capital investments to meet the costs of building information technology infrastructure, procuring innovative technologies and training employees. Such investment is necessary to drive enterprises to achieve business process optimization, improve productivity, expand market share, and enhance innovation. However, the introduction of "the Green Credit Guidelines" tends to exacerbate the financing constraints of heavy polluters [14], which in turn may hinder their active participation in digital transformation. Firstly, from a financial perspective, the financial requirements for digital transformation are usually large, including but not limited to the updating of IT infrastructure, the construction of big data analytics platforms, the introduction of artificial intelligence technologies and related training costs. Heavily polluting enterprises usually face higher environmental risks, and from the "principal-agent cost theory" and "modern contract theory", it can be seen that the principal-agent cost between the bank, as a creditor, and the enterprise will increase with the increase in project risks, including the costs of identification, monitoring, management and auditing. The cost of identification, monitoring, management and auditing, etc. will lead banks to adopt a more conservative strategy when considering costs and benefits. Meanwhile, according to the "risk compensation theory", in order to compensate for the potential environmental risks and possible default risks in the future, banks and financial institutions may require heavy polluting enterprises to pay higher financing costs or put forward more stringent lending conditions [15], such as higher interest rates or additional collateral, in order to obtain the price of risk-bearing compensation. This will lead to higher financing costs for heavy polluters [16]. This means a tighter financial situation for heavy polluting enterprises who are already under pressure to make environmental improvements, reducing their ability to invest in digital transformation.

Secondly, from the perspective of environmental protection and governance costs, the environmental regulatory effect brought about by "the Green Credit Guidelines" will increase the rectification efforts of heavy polluting enterprises to reduce pollution and emissions, which will to some extent reduce the priority and capital investment in digital transformation projects, thus slowing down the process of digital transformation. On the one hand, heavy polluting enterprises may need to reallocate resources in order to comply with the requirements of "the Green Credit Guidelines", which means that enterprises may need to invest more R&D funds and human resources into the end-of-pollution treatment [17], reducing the allocation of funds and resources in digital transformation. This not only makes digital transformation projects significantly less economically attractive within enterprises, but also further inhibits the pace of transformation in the digital field for heavily polluting enterprises. On the other hand, the process of environmental protection management may involve changes such as re-planning of production lines, optimization of production processes, and upgrading of environmental protection facilities. This not only requires the investment of a large amount of resources, but also may lead to disruptions and uncertainties in the production process, bringing additional disturbances to the normal operation of the enterprise. Accordingly, the author proposes the following hypothesis:

H1: "The Green Credit Guidelines" significantly inhibit the digital transformation of heavy polluters.

The amount and quality of an enterprise's R&D investment is directly related to its innovative capacity and future development potential. In today's competitive market, firms that are able to increase their R&D investment on a sustained basis are usually more likely to be able to adapt to market changes and meet future challenges. High levels of R&D investment may play a key role in the digital transformation of heavily polluting firms in weakening the disincentive effect of green credit policies. Firstly, increased R&D investment can make firms more technologically innovative [18], accelerate their digital transformation process, and promote the adoption of more advanced digital technologies. This not only improves productivity and product quality, but also helps to reduce environmental emissions, thus meeting the expectations of green credit policies on environmental requirements. Technological innovation makes enterprises more flexible in digital transformation and allows them to better respond to the environmental standards of the policy, thus weakening the inhibiting effect of the policy on digital transformation. Secondly, a high level of R&D investment helps to improve the productivity of enterprises, and through the application of digital technology, enterprises are able to manage and utilize resources more effectively. Initiatives such as optimizing the supply chain and implementing smart manufacturing can reduce the waste of energy and raw materials and lessen the burden on the environment. This efficient use of resources makes it easier for firms to adapt to the environmental requirements of the policy, diminishing the constraints of green credit policies on digital transformation. Once again, increased investment in R&D demonstrates a firm's commitment to innovation and sustainability. This strategic shift makes firms more inclined to adopt digital technologies to improve productivity and reduce environmental impacts. For heavily polluting firms, digital transformation is not only a technological upgrade, but also a necessary tool to comply with the SDGs. Investments in research and development lead companies towards a digitalization path that is consistent with green credit policies, slowing down the disincentive effect of the policies.

At the same time, investment in R&D is not only about technical aspects, but also includes investment in training and culture. By improving employees' understanding and ability to apply digital technologies, companies can better adapt to the level of technology required for digital transformation and more easily comply with green credit policies. Building green awareness and a culture of sustainability can help firms better integrate digital technologies

and mitigate the disincentive effect of policies on digital transformation. In addition, the relationship between R&D investment intensity and enterprise survivability shows a "U" non-linear relationship, i.e., R&D investment intensity can greatly improve the survivability of enterprises after reaching a certain level [19]. This implies that a moderate increase in R&D investment by enterprises in the process of digital transformation can improve their competitive position in the market while increasing their innovation ability, and mitigate the potential inhibitory effect of green credit policy on their digital transformation. Overall, corporate R&D investment may affect corporate digital transformation on multiple levels by driving technological innovation, improving productivity, promoting sustainable development, and fostering corporate culture. Efforts in all these areas can help weaken the inhibitory effect of green credit policies on the digital transformation of heavy polluting enterprises and enable them to carry out their digital transformation more smoothly. Accordingly, the author proposes the following research hypothesis:

H2: Firms' R&D investment weakens the dampening effect of "the Green Credit Guidelines" on the digital transformation of heavily polluting firms.

The digital transformation of an enterprise is inherently a high-risk business investment, as it involves huge capital investment in new technologies, systems, training and human resources, and such high-cost, resource-intensive investment poses a greater financial challenge to the enterprise. Importantly, digital transformation is usually characterized by greater uncertainty, with technology risk being a key consideration. The introduction of new technologies may lead to technology integration issues and additional costs, and the results and rewards of digital transformation usually take longer to become apparent. In addition, digital transformation requires a cultural shift within the organization, including employee training and adaptation to new ways of working, and this cultural change can be a complex and time-consuming process. Top echelon theory suggests that executives with a financial background typically have a greater tolerance for risk. This trait may have a significant impact in the project decision-making process, making executives more willing to take risks and thus increasing the likelihood that firms will choose riskier projects [20]. Because executives with a financial background typically have a deeper understanding of national policies, market volatility, and financial instruments, they may be more responsive to financial incentives in green credit policies. Compared to their counterparts with non-financial backgrounds, they may be able to utilize green credit resources more effectively in digital transformation and reduce the cost of corporate finance, which in turn will make them more confident in dealing with potential risks, and thus more willing to choose higher-risk investments in corporate projects, leading to a smooth digital technology transition.

At the same time, as executives with a financial background usually have profound financial knowledge and risk management skills, they have a deeper understanding of financial feasibility and benefit analysis. Therefore, they pay more attention to the financial feasibility of enterprise digital transformation in the decision-making process, which helps to establish a more efficient financial review and decision-making process [21], and can more accurately assess the positive impact of green credit policies on the enterprise's financial position compared to others. This financial sensitivity makes them more capable of reducing potential uncertainties through rational financial strategies, and more able to increase enterprises' acceptance of digital transformation, thus more actively promoting enterprises to follow the path of green transformation. Additionally, executives with financial background can use their own financial network resources to establish bank-enterprise contacts, broaden financing channels, reduce the information asymmetry between the enterprise and the bank, so that the enterprise can obtain more funds to alleviate the degree of enterprise financing constraints [22], and further reduce the financial difficulty of digital transformation. Based on the above analysis, the author puts forward the following research hypotheses:

H3: Executive financial background weakens the dampening effect of "The Green Credit Guidelines" on digital transformation of heavily polluting firms.

## Research design

**Model building.** "The Green Credit Guidelines" issued in 2012 provide a good quasi-natural experiment to study the impact of green credit policies on the digital transformation of manufacturing industries. According to the characteristics of this policy, heavily polluting firms should be affected firstly because they face higher environmental risks. Therefore, this paper includes heavily polluting enterprises in the experimental group and non-heavily polluting enterprises in the control group.

Firstly, the following DiD model is constructed to test the impact of green credit policies on the digital transformation of manufacturing industry:

$$Digit_{i,t} = \alpha_0 + \alpha_1 \times Treat_i + Control_{i,t} + \Sigma Year + \Sigma Code + \varepsilon_{i,t} \tag{1}$$

Where "i" stands for firm and "t" stands for time. $Digit_{i,t}$ refers to the level of digital transformation of firm "i" in year "t". $Treat_i$ refers to the treatment group equal to 1, and 0 otherwise. $Post_t$ refers to 1 after the implementation of "the Green Credit Guidelines", and 0 otherwise. $Control_{i,t}$ refers to a set of control variables. $\Sigma year$ is year fixed effects, $\Sigma code$ is individual fixed effects, and $\varepsilon_{i,t}$ is the residual term.

Secondly, in order to test whether the moderating effect of R&D investment between green credit policies and digital transformation of heavy polluting enterprises in the previous hypothesis is valid, the interaction terms of corporate R&D investment and green credit policies, and the interaction terms of executive financial background and green credit policies are introduced into the regression on the basis of model (1) respectively, and the model is set as follows:

$$\begin{aligned} Digit_{i,t} = {} & \beta_0 + \beta_1 Post_t \times Treat_i + \beta_2 Researchinput_{i,t} \\ & + \beta_3 Post_t \times Treat_i \times \text{R}esearchinput_{i,t} + Control_{i,t} \\ & + \Sigma Year + \Sigma Code + \varepsilon_{i,t} \end{aligned} \tag{2}$$

$$\begin{aligned} Digit_{i,t} = {} & \varpi_0 + \varpi_1 Post_t \times Treat_i + \varpi_2 Financial\_background_{i,t} \\ & + \varpi_3 Post_t \times Treat_i \times Financial\_background_{i,t} + Control_{i,t} \\ & + \Sigma Year + \Sigma Code + \varepsilon_{i,t} \end{aligned} \tag{3}$$

Among them, Researchinput represents the level of R&D investment of the enterprise, FinBack$_{i,t}$ represents whether the executives of the enterprise have financial background, and Post×Traet×Researchinput$_i$ and Post×Traet×Researchinput$_i$ represent the interaction between the level of R&D investment of the enterprise and the financial background of the executives, respectively.

**Data sources.** This paper takes listed companies in China's manufacturing industry from 2008 to 2022 as the initial sample, and in order to improve the data quality and ensure the validity of the empirical analysis, the initial sample [23] is screened in accordance with the following criteria: (1) exclude companies with financial anomalies during the sample period, such as ST,* ST, and PT; (2) exclude companies that change their industries between heavy polluting enterprises and non- heavy polluting enterprises during the sample period; (3) exclude key data companies with serious missing data; (4) to avoid extreme values interfering with the findings of this paper, all continuous variables are subjected to the upper and lower

1% shrinkage. Through the above screening, the final sample includes 660 companies with a total of 9,345 observations, of which heavy polluting enterprises contain 220 companies and non- heavy polluting enterprises contain 440 companies; the data used in the study come from the CSMAR database, the iFind database, the Wind database, the National Bureau of Statistics, and MarkData.com, among others.

**Variable selection.** *Explained variable*. The explained variable in this paper is the level of digital transformation of the enterprise, referring to the research results of Chen et al. (2021) [24]: Based on the statistics of 99 digital-related word frequencies in four dimensions: digital technology application, Internet business model, intelligent manufacturing, and modern information system, the digital transformation index of manufacturing enterprises was constructed by using text analysis method and expert scoring method. First, use text analytics to construct Digit_text variables. The first step is to collect the annual reports of listed companies in the manufacturing industry from 2008 to 2022 and convert them into text format, and then extract the text of the business analysis part through Python. The second step is to extract a certain number of samples of enterprises that have been successful in digital transformation through manual judgment. In the third step, the selected samples were processed by word segmentation and word frequency statistics to screen out high-frequency words related to digital transformation, which can be divided into four dimensions: digital technology application, Internet business model, intelligent manufacturing and modern information system, which suggests that we can construct the digital transformation index of enterprises from four dimensions (see Table 1). In the fourth step, based on the words formed in the third step, the text before and after is extracted from the total sample of listed companies, and the text combinations with high frequency are found. The fifth step is to supplement the keywords on the basis of the existing literature to form the final word segmentation dictionary. In the sixth step, based on the self-built word segmentation dictionary, the Jieba function is used to segment all samples, and the number of keyword disclosures is counted from four aspects: digital technology application, Internet business model, intelligent manufacturing and modern information system, so as to reflect the degree of transformation of the enterprise in all aspects. On this basis, the word frequency data was standardized, and the entropy method was used to determine the weight of each index, and finally the Digit text index was obtained.

Secondly, according to the description of the above keywords in the annual report, the number of disclosures, and the production and operation of the enterprise, the expert scoring method is used to judge the degree of digital transformation of each company. Specifically, if "digitalization" is the main investment direction of the enterprise in the year, or "digitalization" has been integrated into the main business of the enterprise (including production, operation, R&D, sales and management, etc.), the Digit_score variable is scored with 3 points; If the enterprise's relevant investment involves "digitalization", but "digitalization" is not the main investment direction at this stage, or the company's main business has not yet achieved deep integration with "digitalization", 2 points will be scored for the Digit_score variable; If the company only touches on a small aspect of "digitalization", or only mentions it in its development strategy and business plan, the Digit_score is set at 1; If there is no mention of "digitalization" in the company's annual report, or if the annual report reflects that the company has not implemented digital transformation, the Digit_score score is 0.

Finally, on the basis of the obtained Digit_text and Digit_score, the final total index Digit is synthesized according to the weight of 50% each, so as to fully reflect the degree of digital transformation of manufacturing enterprises.

*Explanatory variable*. Based on the principle of DID model, the explanatory variable is the interaction "Post*Treat" (DID) of the policy dummy variable (Post) and the industry dummy variable (Treat). Since "The Green Credit Guidelines" came into effect on 24 February 2012,

**Table 1. Enterprise digital transformation index construction and keyword selection.**

| Dimension | Categorical terms | Combinations of text that appear more frequently | Word Segmentation Dictionary |
|---|---|---|---|
| Digital technology applications | Data, numbers, digitization | Data Management, Data Mining, Data Networks, Data Platforms, Data Centers, Data Science, Digital Control, Digital Technology, Digital Communications, Digital Networks, Digital Intelligence, Digital Terminals, Digital Marketing Digitalization | Data Management, Data Mining, Data Network, Data Platform, Data Center, Data Science, Digital Control, Digital Technology, Digital Communication, Digital Network, Digital Intelligence, Digital Terminal, Digital Marketing, Digitalization, Big Data, Cloud Computing, Cloud IT, Cloud Ecology, Cloud Service, Cloud Platform, Blockchain, Internet of Things, Machine Learning |
| Internet business model | Internet, e-commerce | Mobile Internet, Industrial Internet, Industrial Internet, Internet Solutions, Internet Technology, Internet Thinking, Internet Action, Internet Business, Internet Mobile, Internet Application, Internet Marketing, Internet Strategy, Internet Platform, Internet Model, Internet Business Model, Internet Ecology, E-commerce, E-commerce | Mobile Internet, Industrial Internet, Industrial Internet, Internet Solutions, Internet Technology, Internet Thinking, Internet Action, Internet Business, Internet Mobile, Internet Application, Internet Marketing, Internet Strategy, Internet Platform, Internet model, Internet business model Internet ecology, e-commerce, e-commerce, Internet, "Internet +", online Offline, online-to-offline, online, and offline, O2O, B2B, C2C, B2C, C2B |
| Smart manufacturing | Intelligent, intelligent, automatic, numerical control, integration, integration | Artificial intelligence, high-end intelligence, industrial intelligence, mobile intelligence, intelligent control, intelligent terminal, intelligent mobile, intelligent management, intelligent factory, intelligent logistics, intelligent manufacturing, intelligent warehousing, intelligent technology, intelligent equipment, intelligent production, intelligent networking, intelligent system, intelligence, automatic control, automatic monitoring, automatic monitoring, automatic detection, automatic production, numerical control, integration, integration, integrated solution, integrated control, integrated system | Artificial Intelligence, High-end Intelligence, Industrial Intelligence, Mobile Intelligence, Intelligent Control, Intelligent Terminal, Intelligent Mobile, Intelligent Management, Intelligent Factory, Intelligent Logistics, Intelligent Manufacturing, Intelligent Warehousing, Intelligent Technology, Intelligent Equipment, Intelligent Production, Intelligent Networking, Intelligent System, Intelligent, Automatic Control, Automatic Monitoring, Automatic Monitoring, Automatic Detection, Automatic Production, CNC, Integration, Integration, Integrated Solution, Integrated Control, Integrated System, Industrial Cloud, Future Factory, Intelligent Fault Diagnosis, Life Cycle Management, Manufacturing Execution Systems, Virtualization, Virtual Manufacturing |
| Modern information systems | Information, informatization, networking | Information sharing, information management, information integration, information software, information systems, information networks, information terminals, information centers, informatization, networking | Information sharing, information management, information integration, information software, information systems, information networks, information terminals, information centers, informatization, networking, industrial information, industrial communication |

2012 is used as a time dummy variable in this article, and for 2012 and subsequent years, Post is equal to 1, otherwise it is equal to 0. Referring to previous studies [25], this paper selects the Catalogue of Classified Management Industries for Environmental Protection Verification of Listed Companies issued by the Ministry of Environmental Protection in 2008 to identify heavy polluting enterprises, and if they belong to the heavy polluting industries mentioned in the 2008 Ministry of Environmental Protection Notice, they are defined as heavy polluting enterprises. Treat is a grouping dummy variable, with 1 for heavily polluting enterprises and 0 for non-heavily polluting enterprises.

*Control variables*. In order to avoid the estimation bias caused by omitted variables, this paper refers to the results of previous research [26], and selects the following variables as the control variables in the empirical process: (1) Size, (2) Lev, (3) ROE, (4) Tobin Q, (5) Liquid, (6) Cashflow, (7) Loss, (8) Dual.

In summary, the specific definitions of the variables are shown in Table 2.

**Descriptive statistics and analysis.** After the data in this paper were analyzed by descriptive statistics, the results are shown in Table 3. It can be seen that the level of digital

**Table 2. Description of variables.**

| Variable | Variable Description |
|---|---|
| Digit | A total of 99 digital-related word frequencies in the four dimensions of digital technology application, Internet business model, intelligent manufacturing and modern information system were counted, and the digital transformation index of manufacturing enterprises was constructed by using text analysis method and expert scoring method |
| Post | Assign a value of 1 for 2012 and beyond, otherwise 0 |
| Treat | Heavily polluting enterprises are assigned a value of 1, otherwise 0 |
| Size | The natural logarithm of total annual assets |
| Lev | Total liabilities at the end of the year/Total assets at the end of the year |
| ROE | Net Profit/Average Balance of Shareholders' Equity |
| Tobin Q | (Market value of tradable shares + number of non-tradable shares ×net assets per share + book value of liabilities) / total assets |
| Liquid | Current Assets/Current Liabilities |
| Cashflow | Net cash flow from operating activities/total assets |
| Loss | The loss is assigned as 1, otherwise it is 0 |
| Dual | The chairman and general manager are the same person as 1, otherwise 0 |

transformation (Digit) of China's heavy polluting enterprises has a maximum value of 757, a minimum value of 0, and a standard deviation of 42.2630, indicating that there is a large difference in the degree of digital transformation among enterprises. The current ratio (Liquid) has a maximum value of 204.7421, a minimum value of 0.1065, and a standard deviation of 4.4500, indicating that there are also large differences in current ratios among firms. A higher liquidity ratio may indicate a more flexible operation and liquidity, while a lower liquidity ratio may indicate that a company is facing a shortage of funds or assets that cannot be liquidated quickly. Taken together, the descriptive statistics of both the level of digital transformation and the current ratio reveal that there are large differences in the operational management of China's heavy polluters, and that these differences may have an important impact on the competitiveness and long-term development of the enterprises.

## Results and discussion

### Benchmark regression

Table 4 shows the empirical results of the impact of green credit policies on the digital transformation of heavy polluting enterprises, columns (1) and (2) are the cases of regression alone

**Table 3. Descriptive statistics of the main variables.**

| Variable | Observations | Mean | Std. | Min | Max |
|---|---|---|---|---|---|
| Digit | 9345 | 24.5078 | 42.2630 | 0 | 757 |
| Post | 9345 | 0.7506 | 0.4327 | 0 | 1 |
| Treat | 9345 | 0.3386 | 0.4733 | 0 | 1 |
| Size | 9345 | 22.3177 | 1.2580 | 19.0456 | 27.6211 |
| Lev | 9345 | 0.4205 | 0.1861 | 0.0071 | 1.9566 |
| ROE | 9345 | 0.0793 | 0.1425 | -2.1326 | 5.4223 |
| Tobin Q | 9345 | 2.0190 | 1.3769 | 0.6812 | 21.2958 |
| Liquid | 9345 | 2.4106 | 4.4500 | 0.1065 | 204.7421 |
| Cashflow | 9345 | 0.0567 | 0.0705 | -0.5562 | 0.8385 |
| Loss | 9345 | 0.0778 | 0.2678 | 0 | 1 |
| Dual | 9345 | 0.2245 | 0.2245 | 0 | 1 |

**Table 4. Benchmark regression results.**

| | (1) | (2) |
|---|---|---|
| | Digit | Digit |
| DID | -8.257*** | -10.351*** |
| | (-10.113) | (-10.217) |
| Size | | 0.0026*** |
| | | (10.589) |
| Lev | | 0.0002 |
| | | (1.020) |
| ROE | | -0.0002 |
| | | (-1.637) |
| Tobin Q | | 0.0004*** |
| | | (2.909) |
| Liquid | | -0.0003* |
| | | (-1.732) |
| Cashflow | | -0.0001 |
| | | (-0.982) |
| Loss | | -1.305 |
| | | (-1.350) |
| Dual | | 2.791*** |
| | | (3.782) |
| Constant terms | 25.557*** | -16.87*** |
| | (62.405) | (-2.865) |
| Year Fixed Effects | No | Yes |
| Individual fixed effects | No | Yes |
| Number of Observations | 9345 | 9345 |
| $R^2$ | 0.011 | 0.675 |

Note

* * *, * *, * indicate significant at the 1%, 5%, and 10% levels, respectively

Robust standard errors in parentheses, same below.

and adding control variables and fixing the year and individual, respectively. It can be concluded that the DID coefficients are all significantly negative, and the implementation of green credit policies significantly inhibits the digital transformation of heavily polluting enterprises, and hypothesis 1 is verified. The possible explanation is that at present, bank credit is the main financing method for most enterprises in China, and the introduction of the "The Green Credit Guidelines" will make banks more inclined to provide financial support to environmental protection enterprises, while heavy polluting enterprises are difficult to obtain financial support from banks due to serious environmental risks, which will eventually lead to a lack of funds for heavy polluting enterprises, thereby inhibiting a series of technological research and development activities such as digital transformation.

## Robustness check

**Parallel trend test.** To ensure that the results of this paper are not affected by other policies and events, referring to the study of Zhang and Hu (2022) [27], the event study method is used to introduce multiple time dummy variables to construct early and lagged policy variables, and regressions are added while keeping the control variables unchanged. The results of

**Table 5. Parallel trend hypothesis test coefficient distribution table.**

|  | Digit |
| --- | --- |
| Pre4 | 1.4891 |
|  | (3.4019) |
| Pre3 | 0.5147 |
|  | (3.3796) |
| Pre2 | -0.1631 |
|  | (3.2289) |
| Pre1 | 0.5251 |
|  | (3.2168) |
| Current | -20.2375*** |
|  | (3.0982) |
| Post1 | -18.8848*** |
|  | (3.0895) |
| Post2 | -17.4209*** |
|  | (3.0963) |
| Post3 | -14.7533*** |
|  | (3.1114) |
| Post4 | -13.0638*** |
|  | (3.1187) |
| Post5 | -11.0247*** |
|  | (3.1117) |
| Post6 | -9.2666*** |
|  | (3.1272) |
| Post7 | -8.3372*** |
|  | (3.1383) |
| Post8 | -2.4143 |
|  | (3.1415) |
| Post9 | -1.7537 |
|  | (3.1498) |
| Constant | -22.4751*** |
|  | (3.0257) |
| Other Controls | YES |
| Observations | 9345 |
| R-squared | 0.1512 |

the four coefficients before the promulgation of the policy and the coefficients in the last nine periods are shown in Table 5, and the parallel trend test chart is shown in Fig 1, the DID coefficients in the first four periods of the policy are not significant, while the coefficients in the nine periods after the promulgation of the policy are significantly negative. Therefore, the experimental group and the control group are comparable before the implementation of the policy in 2012, and the difference-in-difference regression model in this paper conforms to the parallel trend hypothesis, indicating that the original regression results are robust.

**Placebo testing.** In order to ensure that the impact of "the Green Credit Guidelines" on the digital transformation of heavy polluting enterprises can truly reflect the effect of the policy without being influenced by other factors, drawing on the research results of Guo and Yin (2023) [17], an experimental group is randomly generated to simulate a situation that is not affected by the green credit policy, in order to compare the differences between the experimental group and the control group before the implementation of the policy. This is done by

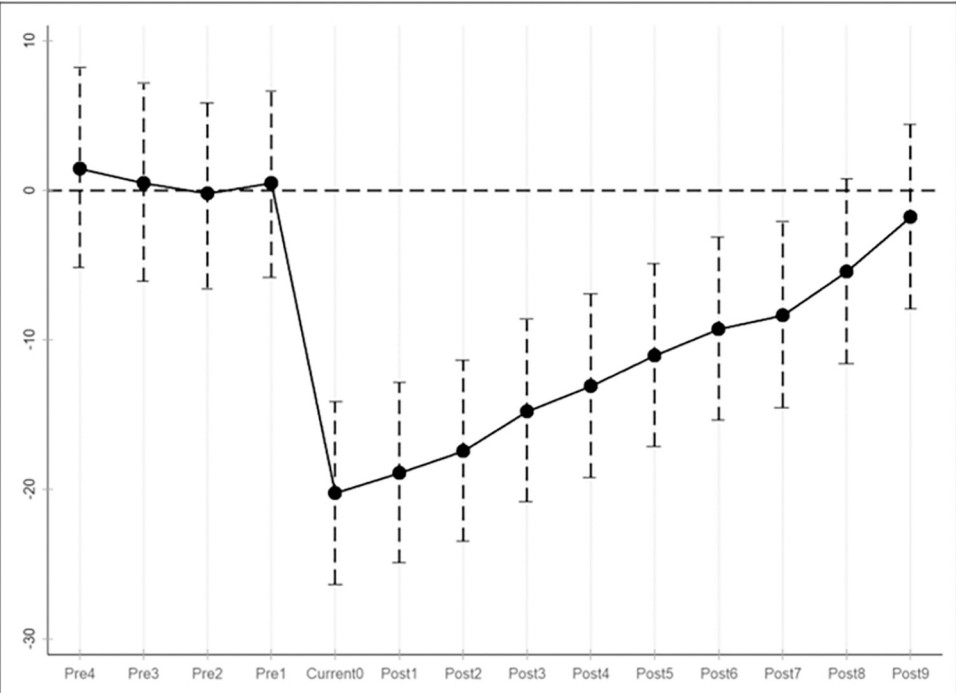

**Fig 1. Plot of parallel trend hypothesis testing.**

randomly, year-by-year and no-putback sampling 2008–2022 enterprises as the experimental group and the rest of the enterprises as the control group, and substituting them into model (1) for regression respectively. The probability density distribution of the coefficient estimates in the placebo test was obtained after 500 random draws and regression tests (see Fig 2). As can be seen from Fig 2, the coefficient estimates from the placebo test are mainly distributed around zero, indicating that the original regression results are robust.

**PSM-DID.** In order to eliminate the endogeneity problem caused by potential selection bias, ensure the robustness of the research results, and improve the comparability of the experimental and control groups in terms of digital transformation, the propensity score matching method was used to conduct the robustness test, drawing on the study of Li (2023) [28]. All control variables in model (1) are selected as matching indicators in the propensity score matching model, and a Logit model is selected to estimate the propensity score, and then nearest-neighbor matching is used to re-match the experimental and control groups to ensure that there is no difference in other factors between the matched experimental and control groups except for the policy differences, and then subsequently re-estimate the model (1). Fig 3 shows that there is a significant difference between the experimental and control groups before matching, and Fig 4 shows the same trend after matching; the DID coefficient is still significantly negative at the 1% level from column (1) of Table 6, which further validates the robustness of the findings of this paper.

**Replacement of core explanatory variables.** This paper replaces the explanatory variables with reference to the research results of Wu et al. (2021) [29], which are statistically derived from a total of 76 digitization-related word frequencies in five dimensions, namely, artificial intelligence technology, big data technology, cloud computing technology, blockchain technology, and the use of digital technology. The regression results are shown in column (2) of Table 6, and the coefficient of DID is still significantly negative, which again verifies the robustness of the findings of this paper.

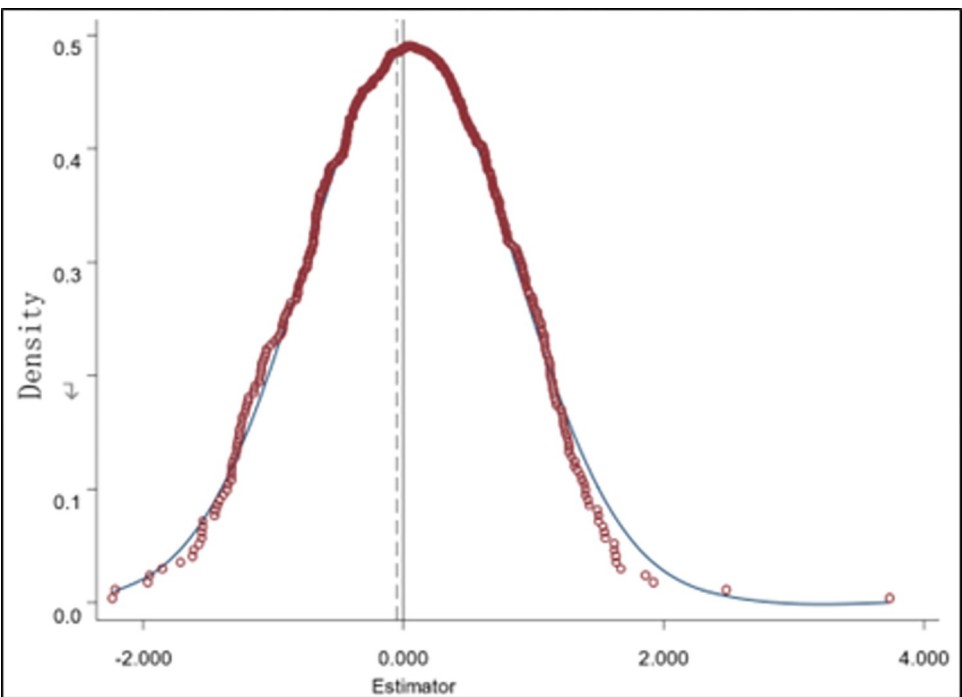

**Fig 2. Placebo test.**

**Treatment of endogenous problems.** Lagging the core explanatory variables by one period helps to alleviate the endogeneity problem and improves the model's ability to explain time correlation and long-term causality. In this paper, by regressing the core explanatory variable DID with one period lag, the results are shown in column (3) of Table 6, and the DID

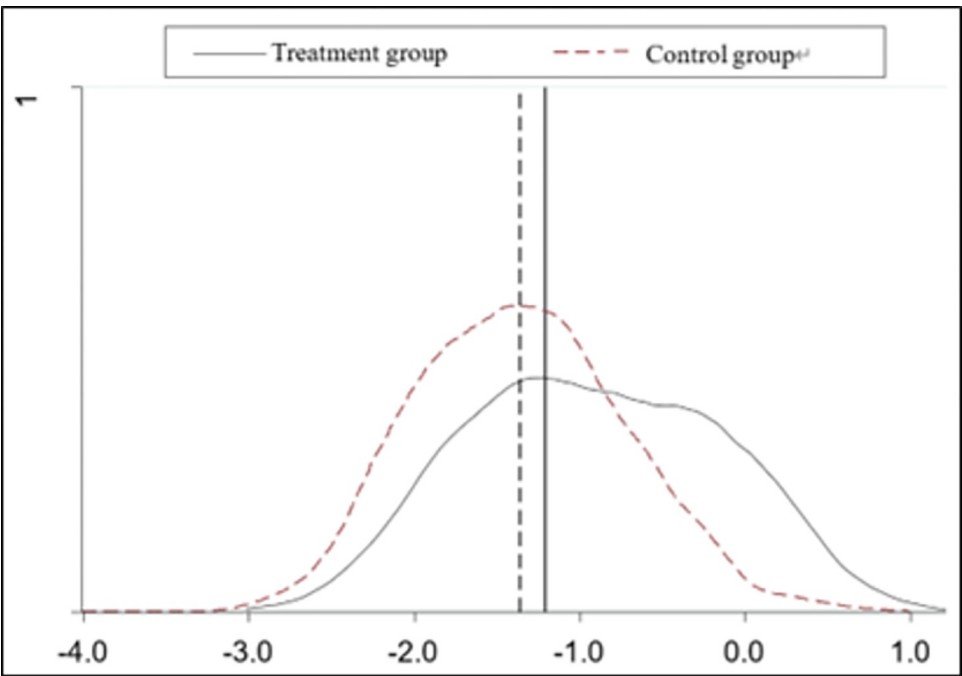

**Fig 3. Density functional plot before matching.**

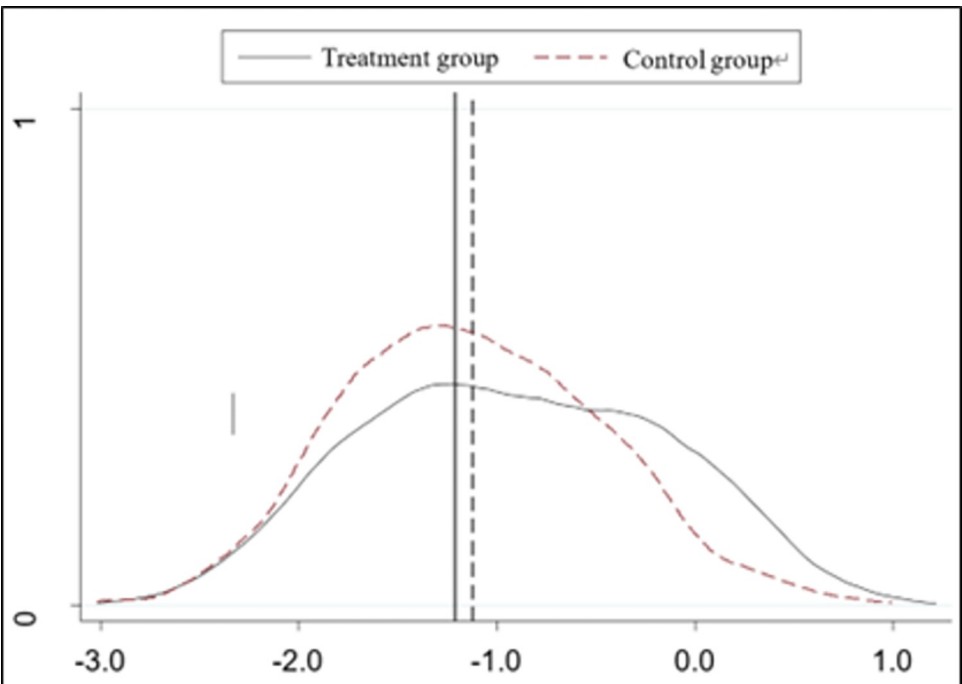

**Fig 4. Density function plot after matching.**

coefficient of is still significantly negative, which indicates that the findings of this paper are still robust after taking into account the time lag effect.

## Mechanism of action tests

**Moderating effects of R&D investment.** The results of the moderating effect test for R&D inputs reported in column (1) of Table 7 show a significantly negative coefficient for R&D inputs and a significantly positive coefficient for the interaction term between R&D inputs and green credit policies. This reflects that overall, enterprise R&D investment weakens the inhibitory effect of the Guidelines on the digital transformation of heavily polluting enterprises, and the inhibitory effect exerted by the policy is more obvious when R&D investment is low, but the inhibitory effect brought about by the policy gradually decreases with the increase of enterprise R&D investment, which suggests that there is a significant substitution

**Table 6. Robustness test.**

|  | (1) PSM-DID | (2) Replacement of core explanatory variables | (3) Core explanatory variables lagged one period |
|---|---|---|---|
| DID | -9.7892*** | -4.6518*** | -10.3512*** |
|  | (-6.9064) | (-11.5656) | (-10.2165) |
| Constant term (math.) | -16.7471** | -6.6295*** | -16.8725*** |
|  | (-2.4916) | (-2.8358) | (-2.8652) |
| Control variable | Yes | Yes | Yes |
| Year fixed effects | Yes | Yes | Yes |
| Individual fixed effect | Yes | Yes | Yes |
| Observed value | 4919 | 9345 | 9345 |
| $R^2$ | 0.6924 | 0.6184 | 0.6749 |

**Table 7. Impact mechanism tests.**

|  | (1) R&D investment | (2) Directors and supervisorsWith financial background |
|---|---|---|
| DID | -0.1383*** | -16.159*** |
|  | (-7.1939) | (-6.045) |
| Researchinput | -0.0614*** | - |
|  | (-6.1617) |  |
| Post×Treat×Researchinput | 0.0283*** | - |
|  | (3.0998) |  |
| FinBack | - | -0.315 |
|  |  | (-0.414) |
| Post×Treat×FinBack | - | 2.817** |
|  |  | (1.984) |
| Constant term (math.) | -0.6130*** | -22.508*** |
|  | (-7.9489) | (-6.091) |
| Control variable | Yes | Yes |
| Year fixed effects | Yes | Yes |
| Industry fixed effect | Yes | Yes |
| Observed value | 9345 | 9345 |
| $R^2$ | 0.2341 | 0.2896 |

relationship between R&D investment and the "the Green Credit Guidelines" in influencing the digital transformation of heavily polluting enterprises, and Hypothesis 2 can be verified. Firstly, the reason why R&D investment can attenuate the inhibitory effect of the green credit policy on the digital transformation of heavy polluting enterprises may be that by strengthening R&D investment, enterprises are more likely to improve their technological level, adopt more environmentally friendly technologies and production methods, and receive more support under the green credit policy, thus alleviating the policy's restriction on the funds required for digital transformation. At the same time, it may indicate that policymakers recognize and encourage firms that meet their environmental goals through independent R&D, as these firms are more likely to succeed in digital transformation; second, the disincentive effect of the policy is relatively more pronounced when R&D inputs are low, which may be due to the fact that the policy puts more emphasis on promoting the digital transformation of firms through financial support, whereas, in the case of low R&D inputs, firms may be more rely on the green credit support provided by the government; finally, the inhibitory effect brought by the policy gradually decreases as the R&D investment of enterprises increases, which suggests that there is an obvious substitution relationship between the R&D investment and the green credit policy in influencing the digital transformation of heavily polluting enterprises, and the possible explanation is that enterprises may prefer to choose to meet the environmental protection requirements through independent R&D, instead of overly relying on the government's green credit policies.

**Moderating effects of executive financial background.** The test results of the moderating effect of executive financial background reported in column (2) of Table 7 show that the coefficient of executive financial background is negative but insignificant and the coefficient of its interaction term with green credit policies is significantly positive, which suggests that executive financial background weakens the inhibitory effect of "the Green Credit Guidelines" on the digital transformation of heavily polluting firms, and Hypothesis 3 is verified. The possible reasons for this are as follows, the advantage of executive financial background lies in its greater tolerance to the high-risk nature of digital transformation. This is mainly reflected in

**Table 8. Heterogeneity test.**

| | (1) State enterprise | (2) Non-state enterprise | (3) Holding of bank shares | (4) Not holding shares in banks |
|---|---|---|---|---|
| DID | -5.069** | -13.418*** | -0.038 | -15.020*** |
| | (-2.195) | (-4.738) | (-0.019) | (-6.971) |
| Constant term (math.) | -83.563*** | -57.042*** | -64.928** | -70.376*** |
| | (-8.192) | (-4.952) | (-5.070) | (-8.480) |
| Control variable | Yes | Yes | Yes | Yes |
| Year fixed effects | Yes | Yes | Yes | Yes |
| Individual fixed Effect | Yes | Yes | Yes | Yes |
| Observed value | 4091 | 5254 | 936 | 8409 |
| $R^2$ | 0.275 | 0.222 | 0.265 | 0.224 |

the fact that financial expertise makes them more sensitive to the financial incentives of green credit policies and more effective in utilizing green credit resources, thus reducing the cost of corporate financing and increasing the acceptance of digital transformation as a high-risk investment. At the same time, gold executives with financial backgrounds have a deeper understanding of financial feasibility and benefit analysis, which reduces uncertainty through rational financial strategies and pushes enterprises to follow the green transformation path more actively. In addition, their financial contacts help broaden financing channels and reduce financing constraints, further easing the financial difficulty of digital transformation.

## Heterogeneity analysis

**Whether the enterprise is a state-owned enterprise.** In this paper, state-owned enterprises (SOEs) and non-state-owned enterprises (NSOEs) are regressed separately, and the results, as shown in columns (1) and (2) of Table 8, indicate that the inhibitory effect of green credit policy on digital transformation is significantly higher for NSOEs than for SOEs. The possible explanations are as follows: firstly, SOEs and non-SOEs play different roles in China's economic environment, with SOEs usually having easier access to government support and financing, while non-SOEs may be more dependent on indirect financing such as bank loans. Green credit policies may lead banks to be more prudent in approving loans and may place greater constraints on the financing needs of non-SOEs, thus inhibiting their digital transformation process; secondly, green credit policies usually require companies to take more steps in environmental compliance to qualify for loans. Non-state-owned enterprises may need more time and resources to meet these requirements, and thus may face greater resistance in the digital transformation process; finally, state-owned enterprises may enjoy market monopolies or more government support in some cases, which may make them more able to bear the costs of digital transformation. In contrast, non-State-owned enterprises may operate in more competitive market environments and be more vulnerable to green credit policies, as digital transformation requires greater capital investment.

**Whether the enterprise holds shares in the bank.** In this paper, firms holding bank shares and firms not holding bank shares are regressed separately. The results show that the inhibition effect of green credit policies on the digital transformation of non-state-owned enterprises is significantly higher than that of state-owned enterprises. The possible explanations are as follows: firstly, that the green credit policy may impose stricter environmental requirements on heavily polluting firms that do not hold bank shares by strengthening loan approval criteria, thereby limiting their access to funds for digital transformation. In contrast, firms that hold bank shares may be more likely to fulfill the conditions of green credit policies

due to closer relationships with financial institutions such as banks. Secondly, firms with different shareholding structures may adopt different strategies in responding to green credit policies. Firms that do not hold bank shares may be more inclined to adopt a strategy of directly confronting environmental requirements by adapting their production and management practices to reduce environmental impacts, while relatively slowing down the pace of digital transformation. In contrast, firms with bank holdings may be more likely to obtain funding through green credits and thus invest more aggressively in digital transformation in order to adapt to environmental trends.

## Conclusions and policy recommendations

Based on "the Green Credit Guidelines" issued in 2012, this paper selects China's manufacturing A-share listed companies from 2008 to 2022 as the research sample. Based on the existing research, this paper uses the DID method to investigate and evaluate the impact of green credit policies on the digital transformation of heavily polluting enterprises. The research results show that: Firstly, the green credit policy, represented by "the Green Credit Guidelines", has a significant inhibitory effect on the digital transformation of heavily polluting enterprises. Secondly, from the perspective of the adjustment mechanism, the R&D investment and the financial background of senior executives weaken the inhibition effect of "the Green Credit Guidelines" on the digital transformation of heavily polluting enterprises, and when the R&D investment is low, the inhibitory effect of the policy is more obvious, but with the increase of enterprise R&D investment, the inhibitory effect of the policy gradually decreases, that is, the R&D investment of enterprises and the Guidelines have an obvious substitution relationship in affecting the digital transformation of heavily polluting enterprises. Thirdly, "the Green Credit Guidelines" has a significantly stronger inhibitory effect on the digital transformation of non-SOE heavy polluting enterprises than that of SOEs; it has a significant inhibitory effect on the digital transformation of heavy polluting enterprises that do not hold shares in a bank, while the effect on heavy polluting enterprises that hold shares in a bank is insignificant.

Based on the above conclusions, this paper puts forward the following policy recommendations from the perspectives of government and enterprises.

On the one hand, the government should launch a special digital transformation loan program to provide heavily polluting enterprises with preferential conditions such as low interest rates and extended repayment periods, so as to ensure that they receive adequate financial support in the process of digital transformation. At the same time, the government should encourage enterprises to increase R&D investment, such as through tax incentives and scientific research funding support, to encourage enterprises to increase R&D investment in the field of digitalization. Flexibly adjust the green credit conditions according to the level of enterprise R&D investment, and provide more flexible credit support for enterprises with low R&D investment. In addition, the government should implement differentiated green credit policies. Formulate differentiated policies according to the nature and shareholding of enterprises, and promote close cooperation between non-state-owned enterprises and non-bank shares and financial institutions to ensure that these enterprises can obtain favorable financial support. On the other hand, enterprises should actively apply for the government's digital transformation loan program to take advantage of low interest rates and flexible repayment terms to reduce financing pressure and ensure the funds needed for digital upgrading. At the same time, enterprises should increase R&D investment and increase digital technology R&D and innovation activities to improve their competitiveness. In addition, enterprises should pay attention to financial literacy training such as digital literacy of senior executives, and encourage enterprises to participate in training programs to enhance their understanding and

support for digital transformation. Finally, companies should optimize their financing structures and strengthen financial cooperation. Specifically, non-state-owned enterprises should explore flexible financing methods and establish close cooperation with financial institutions to obtain favorable financial support. Companies with bank stakes should optimize their financing structures and leverage their banking relationships to obtain better financing conditions to support digital transformation.

## Supporting information

**S1 Data.**
(XLSX)

## Acknowledgments

We would like to express my sincere thanks to the editors and reviewers of the magazine. Thank you for your meticulous review of my manuscript and your valuable comments during your busy schedule. Your professional insights and constructive suggestions have greatly improved the quality and scientific of this paper, and provided important guidance for the refinement and improvement of this study. We have benefited greatly from your hard work and patience in the course of my research. We know that your valuable time and energy play an important role in advancing academic research and knowledge. Therefore, we would like to express my heartfelt respect and gratitude to you for your selfless dedication.

Thank you again for your attention and support to my manuscript, and look forward to your continued guidance and help in the future.

## Author Contributions

**Conceptualization:** Xuan Zhou, Zhengwei Geng.

**Writing – original draft:** Dejia Yuan.

**Writing – review & editing:** Xuan Zhou.

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
