## [Decision Letter · Decision Letter 0]

13 May 2024

PONE-D-24-12713Can Green Credit Policies Improve the Digital Transformation of Heavily Polluting Enterprises: A Quasi-Natural Experiment Based on Difference-in-DifferencePLOS ONE

Dear Dr. Yuan,

Thank you for submitting your manuscript to PLOS ONE. After careful consideration, we feel that it has merit but does not fully meet PLOS ONE’s publication criteria as it currently stands. 

Following peer review and my own assessment, we do not consider your paper to meet the journal’s criteria for publication. We have particular concerns about the underlying hypothesis for the design of the research and the robustness of the methods and analysis.

On the other hand, the referees have suggested the presentation of the work lacks the clarity and intelligibility required.  We strongly recommend you consider the structure of the paper and seek the support of a proofreader or language editor. Therefore, we invite you to submit a revised version of the manuscript that addresses the points raised during the review process.

We look forward to receiving your revised manuscript.

Kind regards,

Juan E. Trinidad-Segovia, PhD

Section Editor

PLOS ONE

Journal Requirements:

3. We note that your Data Availability Statement is currently as follows: All relevant data are within the manuscript and its Supporting Information files

Reviewers' comments:

Reviewer's Responses to Questions

**Comments to the Author**

1. Is the manuscript technically sound, and do the data support the conclusions?

Reviewer #1: Yes

Reviewer #2: Yes

2. Has the statistical analysis been performed appropriately and rigorously? 

Reviewer #1: Yes

Reviewer #2: Yes

3. Have the authors made all data underlying the findings in their manuscript fully available?

Reviewer #1: Yes

Reviewer #2: Yes

4. Is the manuscript presented in an intelligible fashion and written in standard English?

Reviewer #1: Yes

Reviewer #2: No

5. Review Comments to the Author

Reviewer #1: I sincerely appreciate the opportunity to review your work titled "Can Green Credit Policies Improve the Digital Transformation of Heavily Polluting Enterprises: A Quasi-Natural Experiment Based on Difference-in-Difference." I find the topic of your study both interesting and significant. The study focused on the effects of green credit policy on the digital transformation of heavily polluting enterprises. The study revealed that green credit policy significantly inhibits the digital transformation of heavily polluting enterprises. However, in its current state, I believe the manuscript requires further improvements before it can be considered suitable for publication. I have several comments that need to be addressed, along with some thoughts on your findings.

There are fundamental concerns about this study:

1. Clarify the contribution of the study: While the authors briefly mention the key findings of your study, it is important to clearly state the contribution of this study to the existing literature.

2. Identification Strategy: For the results of the event study method, I suggest reporting specific coefficients and reference periods.

3. The conclusion section should offer more profound insights into the conclusions and recommendations.

4. The conclusion section should compare the authors’ results with other papers so that differences and similarities are highlighted.

5. The work is interesting and valuable; however, this work needs to be improved for publishing in this prestigious journal.

6. Regarding writing: The text of this manuscript needs thorough revision, as there are multiple instances of word usage errors, such as "double-difference approach."

Reviewer #2: This paper focuses on the impact of the green credit policy on the digital transformation of heavily polluting enterprises and analyzes the moderating role of enterprises' R&D investment and executives' financial backgrounds. My main concerns are as follows:

1. In the abstract section, both R&D investment and executives' financial background weaken the inhibitory effect, but only the specific impact of R&D investment is described in depth. Why not explain the impact of executives' financial backgrounds? It is suggested to supplement the detailed influence of executives' financial backgrounds or delete the specific expression of the impact of R&D investment.

2. In the introductory section, on page 2 of the manuscript, the first paragraph proposes that environmental protection and digital transformation are the two major themes leading the future development, but only the role of digital transformation on corporate strategy is introduced later, and the impact of environmental protection on corporate strategy is not introduced separately. It is suggested that "on the other hand" should be followed by a description of the importance of environmental protection for manufacturing enterprises. The relevant descriptions of existing green credit policies are suggested to be moved after the introduction of the two themes and introduced as a policy tool linking the two themes. The symbols cited in pages 2-3 of the manuscript are incorrectly marked and should be replaced with superscripts.

3. In the research design section; on page 7 of the manuscript, please re-edit the three model formulas according to the publication standards of the journal, especially formula (3) is deformed. On page 8 of the manuscript, there are errors and repetitions in the description of assignment in the explained variable part of variable selection. There is no mention of how Digit_text calculates, and readers cannot understand the formation process of the explained variable according to the description. On page 9 of the manuscript, there are many repetitions and errors in the wording of the control variables; some variable names in Table 1 are not translated properly, such as "two jobs in one".

4. In the empirical results and analysis section, on page 10 of the manuscript, the results reported in Table 3 are incorrect, possibly missing the lower part of the regression results. On pages 14-15 of the manuscript, the moderating effect of R&D investment and executives with financial background is to test hypothesis 2 and hypothesis 3 respectively, which are incorrectly written as hypothesis 3 and hypothesis 4 in the manuscript.

5. In general, it is not necessary to write the full name of the author in the cited literature in the paper, but the surname can be written. The full name is used in many places in the paper, and it is suggested to modify it. For example, in line 2 of page 11 of the manuscript, "Zhang Xiaoxiao and Hu Jinyan(2022)" should be changed to "Zhang and Hu (2022)".

6. On page 16 of the manuscript, the acknowledgments should be placed at the end of the paper.

7. On page 17 of the manuscript, section of conclusions and suggestions, the first paragraph of the conclusion is repeated and difficult to understand, please re-describe the research content in concise language; the three paragraphs of the research conclusion can be combined into one paragraph, in which "the Guidelines" in the second and third paragraphs are proposed to be changed to "the green credit policy". Policy suggestions should be put forward from the perspectives of different subjects. The current policy suggestions are all from the government’s perspective and lack enterprises’ subjectivity.

8. There are a lot of grammatical and lexical errors in the paper, so it is recommended to polish it carefully.

6. PLOS authors have the option to publish the peer review history of their article (what does this mean?). If published, this will include your full peer review and any attached files.

Reviewer #1: No

Reviewer #2: No

---

## [Author Response · Author response to Decision Letter 0]

13 Jun 2024

Response to Reviewer1

Dear Reviewer

Thanks for the your valuable comments, we have carefully read and understood your revision suggestions. Here, we are deeply grateful for every question you have raised, and have revised accordingly according to your comments. We believe this will greatly improve the quality of our papers.Responses to each question are indicated in red font as follows:

Reviewer #1：Clarify the contribution of the study: While the authors briefly mention the key findings of your study, it is important to clearly state the contribution of this study to the existing literature.

Response #1：Thank you very much for your valuable suggestions, and we have refined and enriched the research contributions in the literature summary section of this article, with a focus on the main contributions to the existing literature. Specifically,“In summary, digital transformation and green credit policies are key factors in the process of high-quality development of the manufacturing industry in terms of technological innovation, transformation and upgrading. At present, there is a large number of literatures on the digital transformation of the manufacturing industry and green credit policies, but few studies combine the two to explore the relationship between green credit policies and the digital transformation of the manufacturing industry. Therefore, the marginal contributions of this paper may be: Firstly, the uniqueness of the research: This paper may be the first time to deeply explore the relationship between digital transformation in the manufacturing industry and green credit policies, combining these two key areas for research. This research is unique in that it connects the two key themes of digital transformation and environmental policies, filling a gap in the existing literature and providing a new research perspective for the academic community. Secondly, the importance of research to academia and practice: This paper fills the gap in the academic understanding of the relationship between digital transformation and green development in the manufacturing industry, and provides new ideas and methods for solving problems in this field. At the same time, the research results of this paper are of great significance for practice, which can provide useful reference suggestions for China's green credit policies formulation and digital transformation of the manufacturing industry, promote the sustainable development of the manufacturing industry, and promote the development of China's economy in a greener and more innovative direction. Thirdly, the theoretical and empirical contributions of the research: By exploring the impact mechanism of green credit policies on the digital transformation of the manufacturing industry, this paper expands the existing theoretical framework and provides new ideas and perspectives for theoretical research. Besides, this paper provides new empirical evidence based on empirical data, deepens the understanding of the mechanism of green credit policies in the process of digital transformation, and provides strong support for practice in related fields. Fourthly, the potential impact of the research: The research results of this paper are expected to have a profound impact on policy-making and practice. By proposing more effective green credit policies to promote the sustainable development of the manufacturing industry, this paper will help guide the government and enterprises to better formulate policies and strategies, promote the development of China's manufacturing industry in a more digital, green and sustainable direction, and contribute to the realization of high-quality economic development.”

Reviewer #2：Identification Strategy: For the results of the event study method, I suggest reporting specific coefficients and reference periods.

Response #2：Thank you very much for pointing out this problem, and we have given a specific coefficient report table for the event study method in the article(Table 4)，so that we can see more deeply the specific situation of the parallel trend hypothesis test. Specifically：The results of the four coefficients before the promulgation of the policy and the coefficients in the last nine periods are shown in Table 4, and the parallel trend test chart is shown in Figure 1, the DID coefficients in the first four periods of the policy are not significant, while the coefficients in the nine periods after the promulgation of the policy are significantly negative. Therefore, the experimental group and the control group are comparable before the implementation of the policy in 2012, and the difference-in-difference regression model in this paper conforms to the parallel trend hypothesis, indicating that the original regression results are robust.

Table 4 Parallel trend hypothesis test coefficient distribution table

 Digit

Pre4 1.4891

 (3.4019)

Pre3 0.5147

 (3.3796)

Pre2 -0.1631

 (3.2289)

Pre1 0.5251

 (3.2168)

Current -20.2375***

 (3.0982)

Post1 -18.8848***

 (3.0895)

Post2 -17.4209***

 (3.0963)

Post3 -14.7533***

 (3.1114)

Post4 -13.0638***

 (3.1187)

Post5 -11.0247***

 (3.1117)

Post6 -9.2666***

 (3.1272)

Post7 -8.3372***

 (3.1383)

Post8 -2.4143

 (3.1415)

Post9 -1.7537

 (3.1498)

Constant -22.4751***

 (3.0257)

Other Controls YES

Observations 9345

R-squared 0.1512

Reviewer #3：The conclusion section should offer more profound insights into the conclusions and recommendations.

Response #3：Thank you for your valuable suggestions. We have made more profound improvements to the conclusion s and recommendations in the draft, and put forward policy recommendations from the perspectives of the government and enterprises. 

The details are as follows:

 Based on the Green Credit Guidelines issued in 2012, this paper selects China's manufacturing A-share listed companies from 2008 to 2022 as the research sample. Based on the existing research, this paper uses the difference-in-differences method to investigate and evaluate the impact of green credit policy on the digital transformation of heavily polluting enterprises. The research results show that: Firstly, the green credit policy, represented by “the Green Credit Guidelines”, has a significant inhibitory effect on the digital transformation of heavily polluting enterprises. Secondly, from the perspective of regulating mechanism, the R&D investment of enterprises and the financial background of executives weaken the inhibitory effect of “the green credit policy” on the digital transformation of heavily polluting enterprises, and the inhibitory effect exerted by the policy is more obvious when the R&D investment is low, but the inhibitory effect brought about by the policy decreases gradually with the increase of R&D investment of enterprises, i.e., the R&D investment of enterprises and “the green credit policy” have an obvious That is, enterprise R&D investment and “the Green Credit Guidelines” have an obvious substitution relationship in influencing the digital transformation of heavy pollution enterprises. Thirdly, “the Green Credit Guidelines” has a significantly stronger inhibitory effect on the digital transformation of non-SOE heavy polluters than that of SOEs; it has a significant inhibitory effect on the digital transformation of heavy polluters that do not hold shares in a bank, while the effect on heavy polluters that hold shares in a bank is insignificant.

Based on the above conclusions, this paper puts forward the following policy recommendations from the perspectives of government and enterprises.

On the one hand, the government should launch a special digital transformation loan program to provide heavily polluting enterprises with preferential conditions such as low interest rates and extended repayment periods, so as to ensure that they receive adequate financial support in the process of digital transformation. At the same time, the government should encourage enterprises to increase R&D investment, such as through tax incentives and scientific research funding support, to encourage enterprises to increase R&D investment in the field of digitalization. Flexibly adjust the green credit conditions according to the level of enterprise R&D investment, and provide more flexible credit support for enterprises with low R&D investment. In addition, the government should implement differentiated green credit policies. Formulate differentiated policies according to the nature and shareholding of enterprises, and promote close cooperation between non-state-owned enterprises and non-bank shares and financial institutions to ensure that these enterprises can obtain favorable financial support. On the other hand, enterprises should actively apply for the government's digital transformation loan program to take advantage of low interest rates and flexible repayment terms to reduce financing pressure and ensure the funds needed for digital upgrading. At the same time, enterprises should increase R&D investment and increase digital technology R&D and innovation activities to improve their competitiveness. In addition, enterprises should pay attention to financial literacy training such as digital literacy of senior executives, and encourage enterprises to participate in training programs to enhance their understanding and support for digital transformation. Finally, companies should optimize their financing structures and strengthen financial cooperation. Specifically, non-state-owned enterprises should explore flexible financing methods and establish close cooperation with financial institutions to obtain favorable financial support. Companies with bank stakes should optimize their financing structures and leverage their banking relationships to obtain better financing conditions to support digital transformation.

Reviewer #4：The conclusion section should compare the authors’ results with other papers so that differences and similarities are highlighted.

Response #4：Thank you very much for your valuable comments, we understand what you mean, you wish we could highlight the differences and similarities between this article and the conclusions of other studies, thank you again! However, it is not too appropriate to put this part in the conclusion section, and this article has compared the research conclusions of this paper with those of other articles in the literature review section, and also highlighted the importance of this research.

Reviewer #5：The work is interesting and valuable; however, this work needs to be improved for publishing in this prestigious journal.

Response #5：Thank you very much for your recognition of this article, and thank you very much for your comments on the changes to this article, which will be of great help to improve the quality of this article. Thank you again for taking the time to review this article, which we have carefully revised and refined in accordance with your suggestions and hope to publish it in this prestigious journal.

Reviewer #6：Regarding writing: The text of this manuscript needs thorough revision, as there are multiple instances of word usage errors, such as "double-difference approach."

Response #6：We have carefully polished the manuscript and carefully corrected any grammatical and lexical errors that occurred. Thank you for your careful review of this article, and thank you again for your hard work on this article.

Response to Reviewer2

Dear Reviewer

Thanks for the your valuable comments, we have carefully read and understood your revision suggestions. Here, we are deeply grateful for every question you have raised, and have revised accordingly according to your comments. We believe this will greatly improve the quality of our papers.Responses to each question are indicated in red font as follows:

Reviewer #1: In the abstract section, both R&D investment and executives' financial background weaken the inhibitory effect, but only the specific impact of R&D investment is described in depth. Why not explain the impact of executives' financial backgrounds? It is suggested to supplement the detailed influence of executives' financial backgrounds or delete the specific expression of the impact of R&D investment.

Response #1：Thank you very much for the lack of a description of the weakening effect of the executive's financial background in the abstract of this article, and we have added a description of its specific impact in the abstract to make the summary more rigorous. Specifically, " Enterprises with senior financial expertise have a deeper understanding of financial feasibility and benefit analysis, and are more receptive to the high-risk investment of digital transformation, while their financial network resources can help broaden financing channels, reduce financing constraints, and further reduce the financial difficulty of digital transformation.". 

Reviewer #2： In the introductory section, on page 2 of the manuscript, the first paragraph proposes that environmental protection and digital transformation are the two major themes leading the future development, but only the role of digital transformation on corporate strategy is introduced later, and the impact of environmental protection on corporate strategy is not introduced separately. It is suggested that "on the other hand" should be followed by a description of the importance of environmental protection for manufacturing enterprises. The relevant descriptions of existing green credit policies are suggested to be moved after the introduction of the two themes and introduced as a policy tool linking the two themes. The symbols cited in pages 2-3 of the manuscript are incorrectly marked and should be replaced with superscripts.

Response #2：Thank you very much for your valuable comments in the introduction of this article, and I very much agree with and thank you very much for your comments, which greatly improves the citation quality of this article. We have already emphasized the importance of environmental protection for manufacturing companies after the " "on the other hand" " in the introduction, while also moving the description of green credit policy back and introducing it as a policy tool that bridges the two themes. Specifically, "On the other hand, environmental protection is of great importance in today's global economy. Manufacturing companies must comply with increasingly stringent environmental regulations and standards, which is not only a social responsibility for enterprises, but also an important way to achieve sustainable development. Environmental requirements are driving companies to innovate in technology and business models, and to explore new development opportunities. Through R&D and application of environmental protection technologies, enterprises can develop new products and services, and open up new markets, which can not only enable the manufacturing industry to meet the regulatory requirements of the green environment and social expectations, but also improve resource utilization efficiency, reduce operational risks, enhance market competitiveness, and explore innovation and development opportunities. Driven by digitalization and environmental protection, manufacturing enterprises should integrate environmental protection into their strategic planning, promote green transformation, and achieve a win-win situation of economic and environmental benefits.”

 In addition, I have replaced all of the symbols cited the manuscript with superscripts.

Reviewer #3：In the research design section; on page 7 of the manuscript, please re-edit the three model formulas according to the publication standards of the journal, especially formula (3) is deformed. On page 8 of the manuscript, there are errors and repetitions in the description of assignment in the explained variable part of variable selection. There is no mention of how Digit_text calculates, and readers cannot understand the formation process of the explained variable according to the description. On page 9 of the manuscript, there are many repetitions and errors in the wording of the control variables; some variable names in Table 1 are not translated properly, such as 

---

## [Decision Letter · Decision Letter 1]

10 Jul 2024

Can Green Credit Policies Improve the Digital Transformation of Heavily Polluting Enterprises: A Quasi-Natural Experiment Based on Difference-in-Differences

PONE-D-24-12713R1

Dear Dr. Yuan,

We’re pleased to inform you that your manuscript has been judged scientifically suitable for publication and will be formally accepted for publication once it meets all outstanding technical requirements.

Kind regards,

Juan E. Trinidad-Segovia, PhD

Section Editor

PLOS ONE

Additional Editor Comments (optional):

Reviewers' comments:

Reviewer's Responses to Questions

**Comments to the Author**

1. If the authors have adequately addressed your comments raised in a previous round of review and you feel that this manuscript is now acceptable for publication, you may indicate that here to bypass the “Comments to the Author” section, enter your conflict of interest statement in the “Confidential to Editor” section, and submit your "Accept" recommendation.

Reviewer #1: All comments have been addressed

Reviewer #2: All comments have been addressed

2. Is the manuscript technically sound, and do the data support the conclusions?

Reviewer #1: (No Response)

Reviewer #2: Yes

3. Has the statistical analysis been performed appropriately and rigorously? 

Reviewer #1: (No Response)

Reviewer #2: Yes

4. Have the authors made all data underlying the findings in their manuscript fully available?

Reviewer #1: (No Response)

Reviewer #2: Yes

5. Is the manuscript presented in an intelligible fashion and written in standard English?

Reviewer #1: (No Response)

Reviewer #2: Yes

6. Review Comments to the Author

Reviewer #1: The authors satisfactorily replied to my remarks and suggestions, and the paper has considerably improved

Reviewer #2: (No Response)

7. PLOS authors have the option to publish the peer review history of their article (what does this mean?). If published, this will include your full peer review and any attached files.

Reviewer #1: No

Reviewer #2: No

---

## [Editor Report · Acceptance letter]

17 Jul 2024

PONE-D-24-12713R1 

PLOS ONE

Dear Dr. Yuan, 

I'm pleased to inform you that your manuscript has been deemed suitable for publication in PLOS ONE. Congratulations! Your manuscript is now being handed over to our production team.

Kind regards, 

on behalf of

Dr. Juan E. Trinidad-Segovia 

Section Editor

PLOS ONE